# How Long Should GPS Recording Lengths Be to Capture the Community Mobility of An Older Clinical Population? A Parkinson’s Example

**DOI:** 10.3390/s22020563

**Published:** 2022-01-12

**Authors:** Lynn Zhu, Patrick Boissy, Christian Duval, Guangyong Zou, Mandar Jog, Manuel Montero-Odasso, Mark Speechley

**Affiliations:** 1Department of Epidemiology and Biostatistics, Schulich School of Medicine and Dentistry, Western University, London, ON N6A 5C1, Canada; gy.zou@robartsinc.com (G.Z.); mmontero@uwo.ca (M.M.-O.); speechly@uwo.ca (M.S.); 2Department of Research and Academics, Ontario Shores Centre for Mental Health Sciences, Whitby, ON L1N 5S9, Canada; 3Ecological Mobility in Aging and Parkinson’s (EMAP) Research Group, Montréal, QC H2L 2C4, Canada; patrick.boissy@usherbrooke.ca (P.B.); mandar.jog@lhsc.on.ca (M.J.); 4Department of Surgery, Orthopaedics Division, Faculty of Medicine and Health Sciences, Université de Sherbrooke, Sherbrooke, QC J1H 5N4, Canada; 5Département des Sciences de L’activité Physique, Université du Québec à Montréal, Montreal, QC H2L 2C4, Canada; duval.christian@uqam.ca; 6Robarts Research Institute, Western University, London, ON N6A 3K7, Canada; 7Parkinson’s Foundation Center of Excellence, London Movement Disorders Centre, London Health Sciences Centre, London, ON N6A 5W9, Canada; 8Department of Clinical Neurological Sciences, Schulich School of Medicine and Dentistry, Western University, London, ON N6A 3K7, Canada; 9Gait and Brain Lab, Parkwood Institute, London Health Sciences Centre, London, ON N6C 5J1, Canada; 10Division of Geriatric Medicine, Schulich School of Medicine and Dentistry, Western University, London, ON N6A 3K7, Canada

**Keywords:** wearable technology, community mobility, GPS, older adults, Parkinson’s disease, recording technique, sampling length

## Abstract

Wearable global position system (GPS) technology can help those working with older populations and people living with movement disorders monitor and maintain their mobility level. Health research using GPS often employs inconsistent recording lengths due to the lack of a standard minimum GPS recording length for a clinical context. Our work aimed to recommend a GPS recording length for an older clinical population. Over 14 days, 70 older adults with Parkinson’s disease wore the wireless inertial motion unit with GPS (WIMU-GPS) during waking hours to capture daily “time outside”, “trip count”, “hotspots count” and “area size travelled”. The longest recording length accounting for weekend and weekdays was ≥7 days of ≥800 daily minutes of data (14 participants with 156, 483.9 min recorded). We compared the error rate generated when using data based on recording lengths shorter than this sample. The smallest percentage errors were observed across all outcomes, except “hotspots count”, with daily recordings ≥500 min (8.3 h). Eight recording days will capture mobility variability throughout days of the week. This study adds empirical evidence to the sensor literature on the required minimum duration of GPS recording.

## 1. Introduction

Capacity and performance of mobility outside of the home around an individual’s community (community mobility; CM) is associated with quality of life [1,2], social participation [3], cognitive abilities [4] and independence [5,6,7]. It is a multidimensional construct, and is often operationalized by researchers as an individual’s spatial-temporal movement patterns (e.g., trips outside, time outside) [8] and travel behavior (e.g., life space, area size travelled, distance travelled by foot versus vehicle) [9,10,11]. Overall, decreased CM is associated with increased risk of disability [5,6] and death [7]. Therefore, assessing and maintaining CM is often a goal of many public health initiatives and clinical treatment plans [12]. Reliable measurement of mobility in the real-life community setting is a key challenge in health and activity research. 

Wearable global position system (GPS) technology with data recorders allows researchers and clinicians to capture a person’s CM easily and passively in his or her real-life setting over multiple hours and days [9,11,12,13,14,15]. These devices often provide data on numerous CM components, such as trip frequency [16,17], duration [8,18,19], distance [20,21,22,23] and size of activity area [9]. A GPS continuously records data, often at every second. Hence, measuring mobility over multiple days generates an enormous amount of real-time data not attainable using recall methods. CM is a dynamic construct, and data captured on an individual can vary greatly even from one day to the next. Thus, even a highly accurate GPS could produce daily recordings with some level of error relative to the true value. 

Inherent technical issues also compromise the quality of large datasets produced by GPS units [24]. An issue specific to the GPS is that increasing the number of GPS recording days may compromise data quality by introducing more opportunities for data loss [23]. For example, GPS studies typically report loss of data due to low battery, signal loss and participant non-compliance [23,25,26]. Studies using 1 week of data collection have reported 11.2% to 53.3% data loss [10,27]. Large data loss compromises the overall data quality and data yield, resulting in biased study results and reducing study power. 

In general, free-living recordings yielded different amounts of data between participants, even in the absence of any missing data. Despite common data loss issues, researchers have not employed a standard length of recording when using GPS to study CM as recording lengths have seldom been evaluated in research. Set recording lengths have varied from 1 day [23,28] to 1 week [10,19]. Within the same study, recording length can also vary each day [21]. Currently, GPS research often does not account for the potential differences in weekday to weekend CM. Studies that do consider daily variations tend to only have captured one or more weekdays that best represent participants’ mobility along with only one weekend day [29]. However, this approach may be prone to selection bias as participants may choose a weekend day based on convenience or social desirability, rather than a day that is representative of typical mobility. 

Currently, it is unknown whether shorter lengths of recording affect the validity of mobility outcomes captured compared to longer lengths. Our objectives for this study are to compare four CM outcomes obtained by a wearable GPS sensor using different recording lengths and recommend a minimally appropriate recording length for using GPS to measure the community mobility of an older clinical population.

## 2. Materials and Methods

Over 14 days, 70 people with Parkinson’s disease wore the wireless inertial motion unit with GPS (WIMU-GPS; Figure 1) during waking hours to capture their typical day-to-day mobility and weekday and weekend mobility. The inclusion and exclusion criteria have been previously published [11]; (Appendix A).

### 2.1. Equipment

The WIMU-GPS [11,15,30]; (Boissy, Hamel, Brière of Canada; patent id: US2012/0232430A1) is a multipurpose wearable sensor platform, combining 3D inertial measures of motion (accelerometers, gyroscope and magnetometers) with a Sirf 3 GPS receiver [30]. Its small pager size allows the unit to be worn non-intrusively around an individual’s torso or hip using a flexible clip-enclosure strap. Inertial measures of motion were sampled at 50 Hz. The GPS data used were sampled at 1 Hz. 

To mimic real-life wear scenarios and minimize testing effects, we asked participants to wear the WIMU-GPS with as little disruption to their daily life as possible. To capture as much free-living mobility as possible, we did not enforce a standard start and end time for data collection each day. We asked participants to remove the WIMU-GPS device before bed in the evenings to charge the battery, and when bathing, swimming or other activities with water. These approaches were consistent with other GPS studies [11,18]. 

### 2.2. Outcomes of Interest

We focused on common GPS-derived community mobility outcomes captured outside of the home, such as daily total “trip count” (i.e., number of trips taken from home) [16,17], “hotspots count” (i.e., number of hotspots visited) [11,15], “time outside” (minutes) [8,18,19] and “area size travelled” (km^2^) [9,31]. We constructed temporal clusters using time-consecutive positions that are within close proximity to each other, over a specific timeframe [15]. A key reference cluster was the home cluster, as created from the first location recorded (i.e., when the GPS was first turned on and calibrated at the participants’ homes). 

“Trip count” was calculated as the total number of trips outside home per day. Trips were operationalized as when a participant leaves the home cluster and returns at a later time in the same day. “Time outside” was calculated by adding the data points identified not within the home cluster over the total recording period [15]. “Hotspots” are the number of clusters reached during each trip, where an individual has stopped for 3 min or more, over the recording period [30]. It excludes routine traffic light stops and allows the detection of commonly visited places. The “area travelled” was derived from a best-fit ellipse drawn around 95% of the GPS points captured during the recording period [30]. Details on the signal processing of the GPS data to compute these outcomes were previously published [15].

### 2.3. Comparison Methods

Community mobility is a complex and multidimensional construct, with several different components such as distance travelled, time spent outside the home, number of destinations, etc. Each component can be measured in terms of frequency, duration and in some cases, intensity (e.g., walking speed) [8,23]. Components of CM can vary in each individual, hourly, diurnally, weekly and seasonally, for reasons including pain, functional impairment and declining health [7,9], as well as lifestyle effects due to changes in habits, personal events, employment factors, transportation, weekday vs. weekend activities, built environment and weather [32,33,34,35,36,37,38].

Assessments based on only one day’s data cannot fully capture CM variation, and the estimates of a participant’s individual true value (ITV) produced using only one day’s data may produce measurement errors of unknown direction and magnitude. Theoretically, a participant’s ITV is calculated as the average of outcomes measured using a known accurate assessment. This value would represent all known and unknown sources of variation in the CM component. In reality, the ITV is estimated because prolonged recording duration is impractical.

### 2.4. Criterion and Comparison Group Selection

To understand which sampling lengths are the most optimal, we compared mobility outcomes recorded using different sampling lengths against each other and against the criterion ITV. Since patterns of mobility are unique to each individual, we separately assessed mobility outcomes recorded, using different recordings for each individual participant. 

### 2.5. Criterion Group

As per the law of large numbers, the best estimates of true CM outcomes (i.e., closest to the true ITV) are achieved when the number of minutes measured per day and number of days measured per sampling period are both maximized. We estimated proxy ITVs for each individual using the greatest number of sampled days with the longest number of recorded minutes. This allows both intra- and inter-day variations in mobility to be captured. 

Given inconsistent GPS wear time, undefined start and end times for each participant and potential missing data, we first defined the most complete dataset that maximizes the total amount of sampled time. Based on previous literature [24], we aimed to use data from participants who recorded at least seven non-consecutive days, with at least one weekend day. We assessed data from participants with at least seven days of recordings to determine the longest daily recording length achieved (i.e., best available estimates of ITV).

In this study, the longest recorded days for all 70 participants contained 1000 min (16.7 h) of recording. This was achieved by only 5 participants over only 8 days in total (each contributing from 1–3 days of ≥1000 min of recording). A proxy ITV formed using these data would not be able to fully capture variations in mobility over week days and weekends. We also observed only one participant recorded ≥7 days of ≥900 min (15 h) which is not a sufficient sample of participants. As a result, we decided the proxy ITV must be based on fewer than 900 min per day.

Next, we observed 14 participants (20.0% of the sample) to have recorded ≥7 days of ≥800 min (13.3 h) of data. Given that this is the most robust sampling permutation that maximized both the number of participants and the number of days/minutes collected, we used this group as the criterion participant group. Each individual’s mean outcomes captured on days with ≥800 min of data recording were used as their proxy ITV.

Despite the relatively small final participant size, we calibrated the GPS to continuously record data at the high frequency of 1 Hz. This allowed us to create a relatively large dataset of minutes recorded which formed the unit of analysis. Our assumption was that these criterion participants provided the best estimates of the true ITV. The criterion ITV for each participant was the mean outcome from all days with at least 800 min of data. Consequently, the number of days used to calculate the ITVs differed between individuals.

### 2.6. Comparison Groups

Criterion participants also recorded days that were shorter than 800 min. These “non-ITV days” were categorized based on the number of minutes recorded and the number of days of each daily recording length. In total, 19 sampling subgroups were formed based on the day and minute permutations that were recorded but only 10 subgroups contained more than one participant (Appendix A). Sampling lengths with only one participant were excluded from the assessment. We systematically compared errors produced by these shorter recording lengths against the criterion ITVs.

### 2.7. Analysis

Demographic characteristics of the criterion group were compared to the non-criterion group using Fisher’s exact test for categorical variables and the two-sample t-test for number of counts and continuous variables. Statistical significance was declared with two-sided *p* < 0.05. 

### 2.8. Subgroup Comparisons

We assessed four outcomes: “trip count”, “hotspot count”, “time outside” and “area size”. Absolute percentage error calculations for each sampling length subgroup used only the “trip count” recorded for each individual and their ITV “trip count”. The criterion ITV for each participant was the mean outcome from all days with at least 800 min of data. Mean ITV “trip count” was calculated for all criterion participants using the mean of all days with recordings of 800+ minutes. Mean non-ITV “trip count” was calculated using all days with recordings of less than 800 min. We calculated the absolute percentage error for each recording length subgroup using the difference between the mean ITV and mean non-ITV values. We repeated the same steps for the other outcomes.

## 3. Results

### 3.1. Demographics

Table 1 shows the difference between the demographic profiles of criterion participants (n = 14) versus participants who did not fulfill the criterion requirements (n = 56). Overall, greater proportions of non-criterion participants were married or living with a common-law partner. Non-criterion participants were slightly older and experienced greater impacts of PD on their overall quality of life. There was no statistical evidence to suggest significant difference between the full sample and criterion participants (*p* > 0.05).

### 3.2. Number of Days Collected

In total, 205 days of varying length were collected from the 14 criterion participants totaling 156, 483.9 min of data (mean = 763.33 ± 210.03 min per day). The number of days with at least 800 min of data contributed by each participant towards their criterion ITV ranged from 7 to 13 days, totaling 113, 466 min of data (minutes = 872.82 ± 73.8 per day). 

### 3.3. Daily and Day-to-Day Variations

Appendix A shows large variations in daily interpersonal “time outside”, “trip number”, “hotspots number” and “area size” travelled. Days with less than 800 min recorded were observed for all but one participant. Insufficient data were observed for all individuals on Day 14 because the last data collection visit often took place mid-day, so data collection on Day 14 was shortened. Table 2 shows the coefficient of variation for each outcome’s ITV days ranging from 39.27% (“time outside”) to 133.44% (“area size”) and indicates that some mobility outcomes were more constant than others for a given individual.

### 3.4. Weekday to Weekend Variations

On average, GPS recordings were 43.15 ± 122.64 min longer during weekend days than weekdays. Although a statistically significant difference in minutes recorded was not observed (*p* = 0.098), longer recordings on weekend days were observed for most of the criterion group (78.6%, n = 14). Figure 2a–d show that mean “time outside”, “trip count”, “hotspot count” and “area size” travelled slightly differed depending on if sampling occurred during the week or on weekends. On weekends, criterion participants tended to decrease their “time outside” by 2.65% (Figure 2a) and the number of hotspots visited by 8.79% (Figure 2c). However, they also were making more frequent and further trips outside ( 5.48% more trips, Figure 2b; 59.25% increase in area travelled, Figure 2d).

Statistically significant differences in mean weekday versus weekend differences were not found for any of the mobility outcomes (Appendix A). Figure 2 also suggests that underestimations in mean “time outside” (−17.96%), “trip number” (−9.33%), “hotspots number” (−15.80%) and “area size” (−19.66%) occurred when shorter non-ITV days were included for analysis (as “all days”) compared to when only ITV days were used. Although statistically significant differences were not found for any of the mean outcomes, consistent underestimation suggesting shorter recording lengths may introduce a level of error in recorded outcomes.

### 3.5. Mean Community Mobility Outcomes 

Table 2 shows the mean community mobility outcomes observed according to different sampling subgroups. Average recordings on ITV days lasted 872.82 (±73.76) minutes whereas an average non-ITV day produced recordings that lasted 573.57 (±233.59) minutes. Compared to ITV days, shorter sampling lengths produced lower mean daily “time outside”, “trip count”, “hotspot count” and “area size” outside the home. On average, relative to recordings from ITV days, non-ITV recordings captured 124.95 fewer minutes outside the home, 0.5 fewer trips and 2.56 fewer hotspots per day. CV analysis shows “area size” to be the most variable mobility construct.

### 3.6. Comparing Sampling to Criterion: Absolute Comparison

#### 3.6.1. Overall CM Outcomes

Figure 3 summarizes the mean error rates when the mobility outcomes collected at each shorter recording length subgroups were compared to the mean outcomes collected during the criterion sampling rate of 7 days of 800 min. All shorter sampling subgroups yielded larger negative mean percentage errors, corresponding to greater underestimation of the proxy ITVs. 

The longest recording length produced the smallest mean error rate of −39.66% (Figure 3). Increasing the number of days with 700–799 min from 1 to 3 days also produced a 21.97% decrease in overall CM error rate.

#### 3.6.2. Specific CM Outcomes

Sampling lengths affect each CM outcome differently. When the data were organized into subgroups according to different minutes over number of days of recording, only recording groups of one day and three days were formed. The recording length of “700–799 min” was the only length captured over three days. As such, we could not compare the error differences when fewer minutes were captured over three days nor when fewer days of various lengths were captured. Figure 4 shows the percentage errors in mean daily relative to the ITV across outcomes, according to different lengths recorded during one day (n = 14).

Across outcomes, sampling length subgroup yielded negative percentage error rates suggesting consistent underestimation of the ITV, except when “1 day of 300–399 min” was collected for “trip count” data. Recordings of this length produced percentage error rates indicating an overestimation of +23.7%. In general, consistent with the underestimation observed when CM outcomes were aggregated (Figure 3), shorter sampling subgroups tend to underestimate CM outcomes relative to the ITV. 

Within a single sampling day, longer sampling lengths tend to yield smaller error rates across most outcomes. This pattern was again inconsistently observed across the four outcomes. For example, at 300–399 min, “trip count” was overestimated by 23.7%, whereas at 400–499 min, “trip count” was underestimated by 60.77%.

## 4. Discussion

The GPS tracker is a widely used data collection tool in clinical use and in health and activity studies [10,23,29,39]. It is often used to capture patterns of movement across location and time (e.g., trips outside, time outside), as well as travel behavior (e.g., area size travelled, distance travelled by foot). Overall, the usefulness of GPS for research is dependent on the quality of the data sampled, which in turn is affected by the recording length [23]. The nature of the outcomes of interest also dictates how strict the data recording process should be. Missing data are common among GPS studies due to variabilities in GPS signal strength during everyday living. For instance, GPS signal interference could be due to building obstruction, indoor versus outdoor activity, weather or time of day [23,25,26]. Sampling strategies using GPS tracker units need to be representative of the true mobility (i.e., outcomes measured should be close to the hypothetical ITV mean, and under- or overestimated errors should be approximately equal), be short enough to minimize participant burden, attrition and cost (i.e., study size cannot be too large and sampling cannot be too long [39], yet achieve sufficient power (i.e., enough recorded time and participants for representative sample and to meet the assumptions of statistical tests). 

Across all outcomes, variation in measurements made on ITV days were smaller than observed on non-ITV days, which supports the proposal that CM estimates collected on ITV days were more precise than the estimates on shorter non-ITV days. Both subgroup and full sample analysis suggested the degree of variability in outcomes is associated with amount of time sampled. 

### 4.1. Sampling Rate by Outcome

The results of this study showed that shorter daily GPS recordings typically underestimate CM outcomes, and longer daily recordings typically result in smaller percentage error. The decrease in percentage error observed across all outcomes, except “hotspots count” at 500–599 min, suggested 500 min (8.3 h) to be the minimum daily recording length that can decrease error in GPS study protocols. For “time outside”, “hotspots count” and “area size”, 600–699 min of recording may be even better at reducing error. 

Recording length affected each mobility construct differently. The highly variable outcome “area size” appeared to be the most sensitive to shorter recording lengths, based on the mean error rates of −97.18 ± 3.02 (range of −91.94% to 100%). Increasing the daily recording length appeared to reduce the error rate, but effects were small compared to the other outcomes. Increasing the number of days sampled from 1 to 3 also improved the error rate for all outcomes except “area size”. It is possible that a threshold number of days may exist for “area size” but this study was too underpowered to determine it. The high degree of day-to-day variability observed for “area size” may also be unique to the study sample. Participants of this study usually travel outside of the home by car, which can produce a larger range of “area size” travelled than travel on foot. It is likely that this large “area size” range is difficult to approximate using shorter days or fewer days. 

“Trip count” yielded the smallest mean error rate (−43.56%) and increasing the sampling length did not always improve the accuracy in outcome. This was evident by a lack of consistent pattern in error rate observed across the subgroups for this outcome. Across all shorter sampling subgroups, the mean participant “trip count” was 1.19 ± 1.49 trips (versus 1.68 ± 1.40 trips during the ITV days). It is possible shorter daily sampling lengths can still capture the few numbers of trips taken daily.

### 4.2. Variability in Mobility 

Previous research has shown mobility to be a variable construct [40,41,42,43], but day-to-day variability has not been quantified in people living with a movement disorder. This study demonstrated daily variations in “time outside”, “trip count”, “hotspot count” and “area size” in older adults with PD.

Although statistically significant differences in mean outcomes were not found according to the day of the week when participants’ data were aggregated, sizeable variations in individual mean mobility outcomes during weekdays versus weekends were graphically observed (Appendix A). This was consistent with reports of day of week variation in physical activity literature [24,44,45], and trip count recorded on Fridays versus other days [17]. As well, the level of discrepancies in GPS recording of “trip count” [19], “duration of walking trip” [19] and “trip travel time” [27] compared to self-reporting has also been shown to differ between weekday versus weekend. The magnitude of the day-to-day variability in mobility outcomes is not the focus in this study and was not quantified. 

It was unclear if the day of week differences in mobility observed in this study were because of set weekly schedules. If participants organize their mobility patterns according to the day of the week, sampling less than one week can lead to systematic bias in mobility based on the day of recording. For example, an individual may only go outside of the home on Wednesdays but no other days. This study also used a sample of mostly retired older adults with PD, whose day-to-day activity patterns may be different from working individuals. The largest variation in the mobility of retired people may not be between weekdays and weekends, but within each day. As well, medication may work better on some days due to variability in pharmacodynamics with different food intake and experiences of fatigue, among other reasons, and can affect mobility of older adults and clinical populations. 

Typical daily mobility patterns cannot be captured with recordings of less than one week. A review of studies using GPS to study various mobility constructs in adults and children suggested that missing data increased beyond 4 days of recording [40]. However, the number of minutes recorded per day was not considered by the review authors. Given the importance of capturing daily variations in mobility, future studies using the GPS must account for the daily and day of the week variations by recording at least one full week of data. 

Motor symptoms of PD tend to affect individuals’ overall functional mobility [34,46] and are alleviated through medications such as levodopa [47]. Therefore, people with PD often schedule physical activities around different medication times throughout a day. For example, people who take levodopa at regular intervals often would wait until the medication’s peak physiological absorption time (i.e., ON-medication state) to get out of bed and may schedule appointments or activities outside of the house to occur after the ON-medication state during a day. In this population, and in other types of mobility disability, it is also important to determine if mobility fluctuations occur through the day. The analyses reported used aggregate data generated every 24 h cycle, as smaller segments of mobility outcomes were not available. Currently, it is not feasible to compute mobility using custom sampling lengths (e.g., 250 min, 280 min). These reasons limit the ability to evaluate smaller segments of diurnal variability. 

### 4.3. Recommended Recording Length

Optimal recording lengths of the GPS can be recommended according to the outcomes studied. A conservative recommendation based on the results of this study is that daily recordings of less than 500 min should be avoided for “time outside”, “trip count”, “hotspot count” and “area size” travelled per day. Ideally, analyses should be based on days with at least 600 recorded minutes for non-discrete continuous outcomes, such as “time outside” and “area size”.

This study is one of the first to quantify intra-individual variability in “time outside”, “trip count”, “hotspot count” and “area size” travelled per day. Past GPS studies often included one day to one week of recording [8]. Although the most optimal number of recording days remains unclear, this study’s results suggested increasing the number of recorded days likely improved the sampling error observed for discrete outcomes such as “trip count” and “hotspot count”. Interestingly, for continuous outcomes, “time outside” and “area size”, increasing the number of days recorded from one to three slightly increased the error rates by 8.47% and 3.78%, respectively.

Despite this, mobility researchers should aim for at least eight days of GPS recording, especially if they are interested in “trip count” and “hotspot count”. Eight days of recording will capture the day-to-day variability in mobility by including one week of multiple weekend days and weekdays. As well, the first and last recorded days of data collection are often shortened due to study logistics. In spite of this, many GPS studies have not accounted for such interruptions. In this study, no data were recorded on Day 14 for any of the criterion participants because they removed the GPS just prior to the last home visit by researchers on this day. Many participants also altered their mobility patterns on study start days and end days to meet with researchers and comply with other study protocols. Including an extra day will improve the chance that a full week of data collection was completed.

### 4.4. Limitations

Although this study focused on assessing the impact of sampling length on commonly reported GPS outcomes, the WIMU-GPS also provided information on mobility outcomes such as “distance travelled by foot”, “distance travelled by car” and “distance to hotspots”. Similar sampling length assessment will be performed in the future for these mobility outcomes as the findings of this study may affect these outcomes differently.

#### Sampling Subgroup and Sample Size

Sampling subgroups used in this study were composed of a small number of individuals. Participant sizes within sampling subgroups were also unbalanced. Small and unbalanced subgroup sizes also violated assumptions of many statistical tests of comparison, so comparisons were limited to descriptive analysis. 

The attempt to optimize the estimate of the true ITV required excluding many participants who did not provide enough data to join the criterion group. As such, our chosen criterion group may have biased the error values reported. The ITV is a hypothetical ideal based on the assumption that increasing data size decreases the sampling error. It is possible that the ITV could be based on a shorter criterion (e.g., at least 7 days of at least 700 min rather than 800 min). However, some levels of error remained while using both one and three days of 700–799 min, so it is likely that the 800 min criterion is still the most optimal. 

As well, the small number of participants in the criterion group prevented more subgroups from being included. This limited the number of quantitative results available on the impact of increasing the number of days sampled. Future studies using a more robust and balanced number of individuals in each subgroup should be performed to determine whether the pattern of shorter recording lengths leading to underestimation is consistently observed. Both improvements are only possible if the overall study sample size could be increased. The current criterion group was achieved after recording 980 days of data (70 participants × 14 days), which is greater than many previous GPS studies e.g., [21,23,29]. Researchers wishing to improve the subgroup sample sizes may wish to assign participants to predefined sampling subgroups a priori instead of post hoc. 

### 4.5. Strengths

#### 4.5.1. Criterion and ITV

Although analysis using the full sample (n = 70) also showed that degree of variability in outcomes is associated with amount of time sampled, a high CV in outcomes was not indicative of data quality alone as the mobility components were intrinsically variable from day to day. Therefore, error rates relative to ITVs were needed. This study recognized the inter-individual variability in mobility and used one’s own mobility outcomes for comparison. The ITV construction included weekends and weekdays, which also helped to improve the approximation of true mobility. Some assessments of missing data have limited their focus on the number of days failing to meet a set daily recorded minutes criterion [23]. This study extends these approaches by closely examining the impact of the different sources of error (e.g., shorter days, smaller number of days). 

Although the criterion group was composed of only 14 individuals, they contributed 205 days of data used for analyses (totaling 156, 483.9 min level data points). Of this, 150 were used for the ITV group (ranging from 7 to 13 days of data). This study maximized the amount of data included for the criterion by using at least 7 days of at least 800 min. The number of minutes and days recorded likely needed to decrease in order to increase the number of participants eligible for the criterion group. Hence, a sample size of greater than 14 could result in a smaller number of minutes included for analysis.

#### 4.5.2. Two Weeks of Sampling

Some GPS studies have sampled more than 70 individuals but over a smaller number of days [9]. Although other GPS studies have used one week of recording for analysis [9,23,39], it is possible that week to week, or even month to month, variation in mobility patterns may occur due to life events or seasonality. The collection of two weeks of data in this study allowed more data to be included in the overall analysis to reduce the chance of anomalous mobility. As well, two weeks of data collection allowed more data to be included in the ITV and more subgroups to be created. Given the high data loss rate, the number of subgroups created may not be possible to achieve with just one week of data. 

## 5. Conclusions

This study demonstrated GPS sampling lengths directly affect the accuracy of CM outcomes collected. By showing that percentage error rates tend to increase when sampling time length decreases, this study illustrated the importance of using an appropriate minimum sampling length that optimizes feasibility, data quantity and representativeness of real-life mobility. 

Recommendations for the minimum number of monitoring days are available for physical activity trackers such as pedometers, accelerometers and self-report logs [44]. However, no recording length recommendations have been made for the GPS. In order to collect information about “time outside”, “trip count”, “hotspot count” and “area size”, the user of the GPS should ensure that daily recordings used for analysis are not less than 500 min in length. However, CM outcomes were differently affected by shorter sampling per day, so different minimum cut offs according to the outcome may still be needed.

Future studies with larger sample sizes are still needed for recommendations about the optimal number of recorded days. However, it is now reasonable to believe that eight days of GPS recording are needed so the day-to-day variability in CM can be captured after filtering out the atypical mobility at the beginning and end of the study period. Therefore, when possible, eight days of recording should be the minimum target, especially when frequency variables are of interest, such as “trip count” and “hotspot count”.

## 6. Patents

Boissy, P., Hamel, M., Brière, S. (2012). Universal actigraphic device and method of use therefor. US2012/0232430A1.

## Figures and Tables

**Figure 1 sensors-22-00563-f001:**
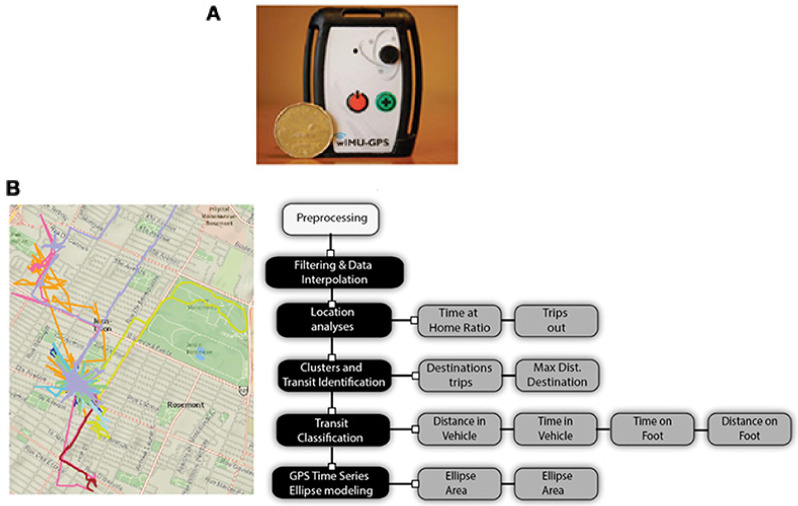
(**A**). Wireless inertial unit with GPS (WIMU-GPS). (**B**). Raw signals and signal processing steps and analysis of GPS signals (previously published in [14]).

**Figure 2 sensors-22-00563-f002:**
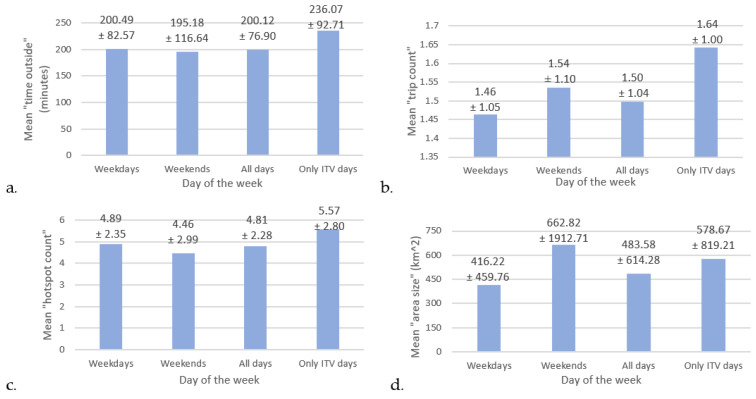
Variations in mean mobility outcomes occurred depending on when sampling occurred during the week and which days were included in the calculations. Mean weekday to weekend change occurred in (**a**). “time outside” (minutes; −2.72% from weekday to weekend), (**b**). “trip count” (+5.19%), (**c**). “hotspot count” (−8.79%) and (**d**). “area size” (km^2^; +37.2%) over the sampling period (n = 14). An increase in mean outcome occurred when only ITV days were used compared to when all days were used, including the shorter non-ITV days (+15.23% for “time outside”, +8.54% for “trip count”, +13.64% for “hotspot count” and +16.43% for “area size”). None of the changes reported for day of the week and type of day included were observed to be statistically significant (*p* > 0.05). (ITV = individual true value; s.d. = standard deviation).

**Figure 3 sensors-22-00563-f003:**
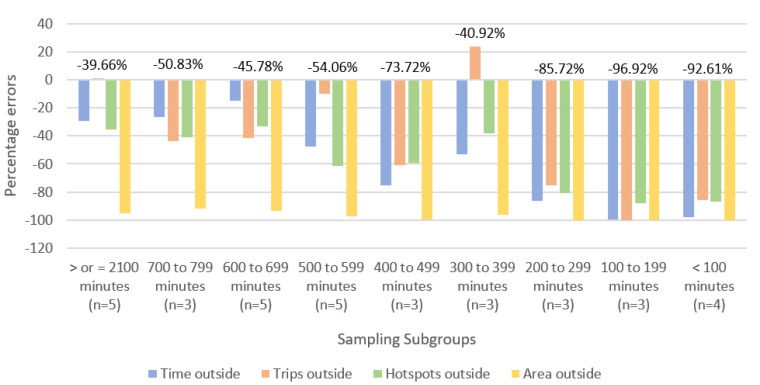
Mean percentage errors for all community mobility outcomes across different sampling lengths relative to sampling length of at least 7 days with 800 or more minutes. Mean percentage error rates for each sampling subgroup are listed above each cluster of bars. All shorter sampling subgroups yielded larger negative mean percentage errors, corresponding to greater underestimation of the ITV. Number of participants in each sampling length subgroup is indicated below the x-axis (n = 14).

**Figure 4 sensors-22-00563-f004:**
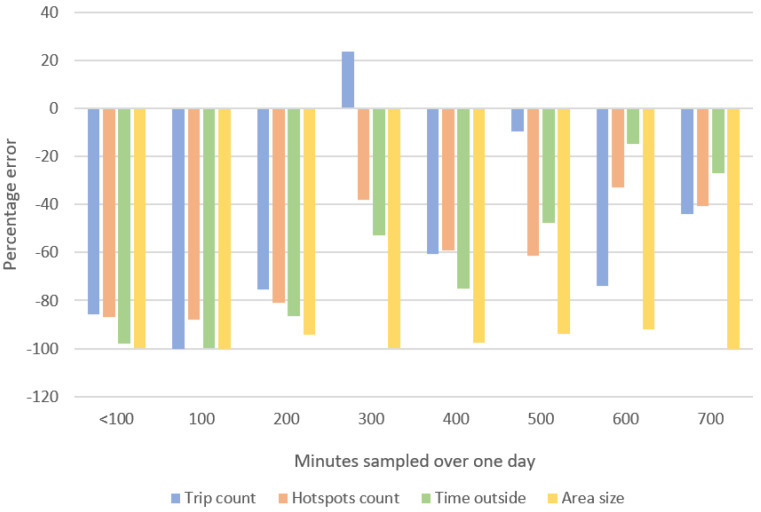
Percentage errors in mean daily, a. “time outside”, b. “trip count”, c. “hotspot count” and d. “area size” recorded during one day (total n = 14). The mean error rates over all subgroups were −58.92 ± 32.12%, −43.56 ± 41.77%, −58.21 ± 22.62% and −97.18 ± 3.02%, respectively.

**Table 1 sensors-22-00563-t001:** Demographic characteristics of the selected criterion participants compared to all participants. (PD = Parkinson’s disease; MoCA = Montreal Cognitive Assessment; PDQ-39 = Parkinson’s Disease Questionnaire—39 items; s.d. = standard deviation).

	Criterion Participants (n = 14)	All Participants (n = 56)
Demographics covariates	Mean ± s.d.; n (range or %)	
Age (years)	69.2 ± 6.5 (57–79)	67.1 ± 6.3 (55–79)
Sex		
Male	8 (57.1%)	39 (69.6%)
Female	6 (42.9%)	17 (30.4%)
Marital status		
Unmarried/widowed/	6 (42.9%)	
separated	8 (57.1%)	7 (12.5%)
Married/common law		49 (84.4%)
Employment status	Fully retired = 12 (85.7%)	Fully retired = 47 (83.9%)
	Partial or full employment = 2 (14.3%)	Partial or full employment = 9 (16.1%)
Residential setting	Urban = 5 (35.7%)	Urban = 12 (21.4%)
	Suburban = 1 (7.1%)	Suburban = 15 (26.8%)
	Rural, in town = 5 (35.7%)	Rural, in town = 19 (33.9%)
	Rural, outside of town = 3 (21.4%)	Rural, outside of town = 10 (17.9%)
Living situation	Alone = 5 (35.7%)	Alone = 8 (14.3%)
	With family/friends = 9 (64.3%)	With family/friends = 48 (85.7%)
Driving status	Drives = 14 (100%)	Drives = 51 (91.1%)
	Does not drive = 0 (0%)	Does not drive = 5 (8.9%)
MoCA	26.6 ± 2.5 (23–30)	25.3 ± 3.0 (18–30)
Time since PD diagnosis (years)	5.4 ± 4.0 (<1–14)	6.4 ± 5.6 (<1–30)
Impact of PD on overall quality of life (PDQ-39 scores; 0 = no impact, 100 = total impairment)	13.9 ± 15.8 (2.1–64.7)	20.8 ± 12.4 (1.8–51.4)

**Table 2 sensors-22-00563-t002:** Average community mobility outcomes recorded using different sampling lengths (n = 14). Days with 800 or more minutes of recording constitute ITV days, and those with less are non-ITV days. Coefficient of variation for ITV days used the mean and s.d. of all ITV days in the formula: s.d./mean∗100. Non-ITV mean values were calculated using the average for all non-ITV subgroups, and this mean was used to calculate the non-ITV coefficient of variation. (ITV = individual true value; CM = community mobility; s.d. = standard deviation.).

CM Outcomes	All Non-ITV Days		ITV Days	
Mobility Outcomes	Mean	Coefficient of Variation (s.d./mean∗100)	Mean	Coefficient of Variation of the ITV (s.d./mean∗100)
Time outside in minutes (range)	119.95 ± 135.34 (0.7–465.02)	112.83%	244.9 ± 169.95 (0.03–712.47)	69.40%
Trip count (range)	1.19 ± 1.49 (0 to 8)	83.31%	1.68 ± 1.40 (0–7)	83.33%
Hotspot count (range)	3.19 ± 2.93 (0 to 16)	78.30%	5.75 ± 4.50 (1–27)	78.26%
Area size in km^2^ (range)	182.68 ± 732.12 (0 to 4241.77)	400.77%	671.63 ± 1758.4 (0–10,250)	261.81%

## Data Availability

Restrictions apply to the availability of these data as they were collected from patients of the National Parkinson’s Foundation Center of Excellence, London Movement Disorders Centre (London Health Sciences Centre, Canada). Current agreements about privacy do not allow us to make these data publicly available.

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
