# Peer review of "How Long Should GPS Recording Lengths Be to Capture the Community Mobility of An Older Clinical Population? A Parkinson’s Example"

_sensors, 2022, doi:10.3390/s22020563_

Round 1

Reviewer 1 Report

I was able to read this paper with pleasure (Manuscript ID: sensors-1502216).

The objectives of the reviewed study are to: compare four community mobility outcomes obtained by a wearable GPS sensor using different recording lengths and recommend a minimally appropriate recording length for using GPS to measure the community mobility of an older clinical population.

The title of this study is relevant to public health. The use of the proposed technology supports the caregivers of elderly people with movement disorders in monitoring their life activity. The mentioned control increases the possibilities of independent life activity of the elderly and improves their quality of life.

GPS technology has been used in the studied problem before , however, the authors of this study made an attempt to determine the minimum length of GPS sampling, which affects the accuracy of the collected results.

The authors very efficiently describe the selected research problem, demonstrating at the same time significant practical experience. The presented conclusions correspond to the aim of the study and are in line with the presented evidence. Professional literature sources were used in the study (47 items, 17% from the last five years).

Recently, there is a lack of studies combining cognitive and utilitarian aspects. The manuscript is attractive and clearly written. The authors very efficiently describe the selected research problem, demonstrating at the same time significant practical experience.

My only comment is on the explanation of the abbreviations below the tables and figures, both in the manuscript and supplementary materials. I know that they were previously explained in the text, but placing them under the graphic will make it easier for the reader to interpret its content.

Best regards,

Reviewer

Author Response

(Please see the attachment for a record of all of the responses.  Thank you.)

REVIEWER 1:

Comments: I was able to read this paper with pleasure (Manuscript ID: sensors-1502216).

The objectives of the reviewed study are to: compare four community mobility outcomes obtained by a wearable GPS sensor using different recording lengths and recommend a minimally appropriate recording length for using GPS to measure the community mobility of an older clinical population.

The title of this study is relevant to public health. The use of the proposed technology supports the caregivers of elderly people with movement disorders in monitoring their life activity. The mentioned control increases the possibilities of independent life activity of the elderly and improves their quality of life.

GPS technology has been used in the studied problem before , however, the authors of this study made an attempt to determine the minimum length of GPS sampling, which affects the accuracy of the collected results.

The authors very efficiently describe the selected research problem, demonstrating at the same time significant practical experience. The presented conclusions correspond to the aim of the study and are in line with the presented evidence. Professional literature sources were used in the study (47 items, 17% from the last five years).

Recently, there is a lack of studies combining cognitive and utilitarian aspects. The manuscript is attractive and clearly written. The authors very efficiently describe the selected research problem, demonstrating at the same time significant practical experience.

Suggestions:

My only comment is on the explanation of the abbreviations below the tables and figures, both in the manuscript and supplementary materials. I know that they were previously explained in the text, but placing them under the graphic will make it easier for the reader to interpret its content.

              Authors’ Response:

Thank you very much for the kind words and suggestions.  We have added the abbreviations in all of the Table and Figure captions for clarity.

Reviewer 2 Report

The work is interesting and the authors has done lots of sound invitations on the topic.

 I make some comments on the attached PDF document, please it and make related revision.

  1. one of my concern is: wear indoor or out door, we know that GNSS is almost useless indoors, and about 80% of the activities are basically indoors, especially patients. how you process this?
  2. what is the true CV? how you identify it  in  line 150 page 4
  3. in conclusion the author should give "Need to give quantitative results".

Author Response

(Please see the attachment to Reviewer' feedback for the cover letter to the Editor and a record of all of the responses. Thank you.)

REVIEWER 2:

Comments: The work is interesting and the authors has done lots of sound invitations on the topic.

I make some comments on the attached PDF document, please it and make related revision.

            Authors’ Response:

Thank you very much for the kind words and the attentive review.  We have reviewed every comment and revised the manuscript accordingly. In particular, thank you for asking about the negative mean percentage errors reported in Figure 3 (Section 3.6.1 CM Outcomes). The negative sign shows that the mean outcomes collected when sampling was shorter tended to underestimate the proxy ITVs. We added this description to the Figure 3 captions.

Suggestions:

  1. one of my concern is: wear indoor or out door, we know that GNSS is almost useless indoors, and about 80% of the activities are basically indoors, especially patients. how you process this?

    Authors’ Response:

    Thank you for this question.  We agree that indoor recordings are often imprecise and that the bulk of a day’s activities tend to be indoors.  As such, we only analyzed and included outcomes captured outdoors that describes one’s mobility in the community (i.e., “trip count”, “hotspots count”, “time outside” and “area size travelled”). We described trips outdoors as having occurred when participants leave a predetermined home reference cluster.  As described in lines 121-123 of the manuscript, the home cluster was created when the GPS was first turned on and calibrated at the participants’ homes.

    For a more thorough description of the processing methods of outdoor data, we have referred readers to a previous study cited in Section 2.2 of the Methods.

    To emphasize that only outdoor outcomes were analyzed, we included “captured outside of the home” in the description of community mobility counts in Section 2.2 (Lines 116 – 117).

  2. what is the true CV? how you identify it  in  line 150 page 4

    Authors’ Response:

    Thank you for noting the need for clarity. The true ITV can only be estimated and in this study, we used a proxy criterion value derived after maximizing both number of days and hours sampled.  In this sentence, we erroneously referred to this as the “true criterion value”.  This has been fixed in Lines 154-153 to say “the criterion ITV”, which is defined in the next paragraph.

    3. in conclusion the author should give "Need to give quantitative results".

                Authors’ Response:

Thank you for this suggestion. The overall conclusion of this study was the recommendations regarding the number of sampled days and minutes that reduced the error rate in the community mobility outcomes recorded. We felt reporting quantitative results about the size of the error rates for each sampling permutation would lack context in the Conclusion section and may not be as useful to readers as the overall conclusion.  

Reviewer 3 Report

In this manuscript, the authors present a study of trying to discover the (optimal) length of the GPS recordings for best quantifying the community mobility (CM) of PD patients including daily “time outside”, “trip count”, “hotspots count” and “area size travelled”. Whilst the CM information is very important for those working with older people and/or patients to monitor and maintain their mobility level, the methodology proposed in the manuscript somewhat lacks a bit of sound scientific basis.  More specifically, the estimation of ITV by using the chosen criterion participant group (those who recorded >=7 days of >= 800 167minutes (13.3 hours) of data) is a bit arbitrary – it is very hard to justify the use of the Law of Large Numbers here. The assumption that these criterion participants provided the best estimate of the true ITV is also very much arbitrary. As a consequence, this chosen criterion results in the bias of all the results (negative error). 

Author Response

(Please see the attachment to Reviewer' feedback for the cover letter to the Editor and a record of all of the responses. Thank you.)

REVIEWER 3:

Comments/suggestions: In this manuscript, the authors present a study of trying to discover the (optimal) length of the GPS recordings for best quantifying the community mobility (CM) of PD patients including daily “time outside”, “trip count”, “hotspots count” and “area size travelled”. Whilst the CM information is very important for those working with older people and/or patients to monitor and maintain their mobility level, the methodology proposed in the manuscript somewhat lacks a bit of sound scientific basis.

 More specifically, the estimation of ITV by using the chosen criterion participant group (those who recorded >=7 days of >= 800 167minutes (13.3 hours) of data) is a bit arbitrary – it is very hard to justify the use of the Law of Large Numbers here. The assumption that these criterion participants provided the best estimate of the true ITV is also very much arbitrary. As a consequence, this chosen criterion results in the bias of all the results (negative error). 

            Authors’ Response:

Thank you very much for the thoughtful comment.

Our study had a high degree of missing data over the sampling period (which is common for GPS studies). As well, community mobility outcomes show a high degree of inter- and intra-variation. Taken together, we did not feel it was appropriate to compare the sample means to ITVs derived from group means, which would capture the most amount of data overall. Instead, we selected our criterion participant group and used their proxy ITV as the comparison because this was the most robust sampling permutation that maximized both the number of participants and the number of days/minutes collected.  

The formation of the criterion participant group was done in a careful stepwise fashion.  This was previously described in Supplemental B Criterion Group Formation. For clarity, we have now moved this Supplemental description to the body of the manuscript (please see revised Section 2.5 Criterion Group, lines 161-187).

We also acknowledged that the selected criterion group may have biased the error values reported in the Limitation section of the Discussion (lines 510-511).

Reviewer 4 Report

General comments

This manuscript aims at comparing four community mobility outcomes obtained by a wearable global navigation satellites system sensor (GPS) using different recording lengths and recommending a minimally appropriate recording length for using GPS to measure the community mobility of an older clinical population. Overall, authors manage to fulfill properly their aims.

Minor comments

(line 31 and elsewhere throughout MS) … a GPS re-… (viz., extra spaces)

(References) something wrong regarding numbering?

(l47 and elsewhere throughout MS) Please, do not start sentences with acronyms;

(l91) Please, add make and country;

(l101) Hz (please, be consistent throughout MS);

(l105) Figure 1 not introduced in text;

Ref #31 not in text;

(l297) please, do not use acronyms in headings;

Ref #46 not in text.

Author Response

(Please see the attachment to Reviewer' feedback for the cover letter to the Editor and a record of all of the responses. Thank you.)

REVIEWER 4:

Comments: This manuscript aims at comparing four community mobility outcomes obtained by a wearable global navigation satellites system sensor (GPS) using different recording lengths and recommending a minimally appropriate recording length for using GPS to measure the community mobility of an older clinical population. Overall, authors manage to fulfill properly their aims.

            Authors’ Response:

Thank you very much for the kind words, thorough review and suggestions.  We have addressed all of the suggestions noted by the reviewer (see below).

Suggestions:

  1. (line 31 and elsewhere throughout MS) … a GPS re-… (viz., extra spaces)

    Authors’ Response:

    Thank you for noting this. The extra spacing noted in this incidence was because the journal template used for this submission automatically justified the alignment of all of the manuscript text, which created unequal spacing between words. We have reviewed the manuscript to delete additional spacing that was not due to formatting.

  2. (References) something wrong regarding numbering?

    Authors’ Response:

    Thank you. We checked the numbering and added previously missing references #31 and 46 in the text (see below).

  3. (l47 and elsewhere throughout MS) Please, do not start sentences with acronyms;

    Authors’ Response:

    Thank you, we have written out the acronym or changed the wording here and throughout the manuscript.

  4. (l91) Please, add make and country;

     Authors’ Response:

            Thank you, we have included the maker name, country and patent id (line
            98-99).

  1. (l101) Hz (please, be consistent throughout MS);

     Authors’ Response:

    Thank you, we have changed it to the correct notation throughout (i.e., lines 103, 190).

  2. (l105) Figure 1 not introduced in text;

     Authors’ Response:

    Thank you for noting this, we have added this in Page 2, Section 2 Materials and Methods, Line 92.

  3. Ref #31 not in text;

     Authors’ Response:

    Thank you.  This reference has been added to line 119 to describe a previous example of using GPS to capture older adults’ area travelled/life space size.

  4. (l297) please, do not use acronyms in headings;

     Authors’ Response:

    Thank you. We have written out the acronym in the Figure 3 captions.  In other figures/tables, we have written out those that were not key concepts.  For the few acronyms that were kept for brevity, we have now added their definitions in the captions.

  5. Ref #46 not in text.

     Authors’ Response:

    Thank you.  This reference has been added to line 459 to describe an updated understanding of how Parkinson’s disease affects the functional mobility in people living with the condition.

Round 2

Reviewer 2 Report

The author replied to my comments well, and i suggested it can be published.

Reviewer 3 Report

The revised version is fine.